# Destiny of Two Caddisfly Species under Global Climate Change

**Katarina Stojanović** [1,*], **Dubravka Milić** [2], **Milica Ranković Perišić** [2], **Marija Miličić** [3] and **Ivana Živić** [1]

[1] University of Belgrade, Faculty of Biology, 11000 Belgrade, Serbia; ivanas@bio.bg.ac.rs
[2] University of Novi Sad, Faculty of Sciences, Department of Biology and Ecology, 21000 Novi Sad, Serbia; dubravka.milic@dbe.uns.ac.rs (D.M.); milica.r@dbe.uns.ac.rs (M.R.P.)
[3] University of Novi Sad, BioSense Institute—Research Institute for Information Technologies in Biosystems, 21000 Novi Sad, Serbia; marija.milicic@biosense.rs
[*] Correspondence: k.bjelanovic@bio.bg.ac.rs

**Abstract:** Climate change is considered one of the greatest threats to freshwater biodiversity. Although freshwater biodiversity is an important contributor to economic, scientific, and cultural aspects of human society, freshwater species, especially invertebrates, tend to be neglected in conservation studies. This fact also raises the question of the suitability of protected areas (PAs) for the conservation of freshwater biodiversity. In our study, we used species distribution models (SDMs) to examine the effects of climate change on the two trichopteran species *Helicopsyche bacescui* Orghidan and Botosaneanu, 1953 and *Thremma anomalum* McLachlan, 1876. We determined which areas in the Balkans and neighboring countries might be lost to or colonized by these species in the future, and tested the effectiveness of PAs for the conservation of freshwater biota. While *H. bacescui* will potentially lose up to 68% of its range, *T. anomalum* could expand its range by up to 72%. Both species tend to shift their range mainly to the Carpathian Mountains. Our results suggest that currently established PAs are insufficient to cover the potential current and predicted future ranges of the studied species. The study therefore highlights the need to combine aquatic and terrestrial systems in the future designation of protected areas.

**Keywords:** caddisflies; global warming; climate change; species distribution models; conservation; protected areas

## 1. Introduction

The diversity of both species and habitats found in freshwater ecosystems is disproportionally high in relation to their small share in the total area of the Earth's surface. Unfortunately, recent studies of biodiversity decline have shown that these ecosystems are experiencing the greatest loss of species [1,2]. More than 15 years ago, Dudgeon et al. [3] listed habitat destruction, overexploitation, exotic species invasion, flow modification, and water pollution as the major factors leading to range restriction and population declines in freshwater ecosystems worldwide. However, in their recent review, Reid et al. [4] identified 12 emerging Anthropocene threats affecting freshwater biodiversity, the first being climate change. Climate change has numerous direct and indirect impacts on freshwater bodies, including temperature increases, variations in annual precipitation, and changes in flow regimes, as well as increases in pathogens and more intense eutrophication [5–7]. The manifestation of climate change through biodiversity loss can be seen at the various levels of biological organization (from genes to ecosystems) and at the various levels of spatial scales (local or regional) [8,9].

Due to the increasing threat of climate change, species distribution models (SDMs) are widely used to predict potential range changes of species as a result of temperature increases [10]. Species distribution models summarise the occurrence of species at specific locations and under the environmental conditions there, allowing the prediction of suitable environments at different geographic ranges or time frames [11]. There are many examples

where SDMs have been used to study the effects of climate change on species distributions, but not as many that deal with freshwater invertebrates [12–16].

Although freshwater biodiversity is an important contributor to the economic, scientific, and cultural aspects of human society [3], freshwater species, particularly invertebrates, are clearly neglected in conservation studies [17]. Only about 10,000 freshwater invertebrates worldwide are listed on the IUCN Red List of Threatened Species to date [18]. Conservation efforts and funding tend to focus on more charismatic species, mainly terrestrial invertebrates [17]. The main obstacle to bringing freshwater species into the focus of conservation action is the lack of data on both their conservation status and distribution [19]. For this reason, better-studied groups usually serve as the basis for the development of conservation measures and designation of protected areas [20]. However, this approach raises the question of how appropriate protected areas (PAs) are for the conservation of freshwater biodiversity [21,22].

In our study, we focus on two Trichoptera species, *Helicopsyche bacescui* Orghidan and Botosaneanu, 1953 (Helicopsychidae) and *Thremma anomalum* McLachlan, 1876 (Thremmatidae). Larvae and adults are easily recognized based on their morphological characteristics [23–25]. Larvae build specific and easily recognizable cases composed entirely of mineral particles of varying sizes. As the name of the genus suggests, *H. bacescui* builds helical cases, while *T. anomalum* has a tubular (horn-like) case that is dorsally bent, with a larger anterior opening. Both species are restricted to headwater streams, in mainly eucrenal to hypocrenal zones [26–28], while *H. bacescui* has also been recorded in small streams with intermittent flows [27]. As substrate types, they prefer pebbles to cobbles and cobbles to boulders [26], and both are grazers [24,25]. These species are considered cold-stenothermic and inhabit waters with temperatures mainly up to 9 °C [26]. Their distribution covers a large part of the Balkan Peninsula (Figure 1), including the following ecoregions: ER 5 (Dinaric Western Balkan), ER6—Hellenic Western Balkan, ER7—Eastern Balkan, as well as ER 10—The Carpathians, ER 12—Pontic Province, ER 24—The Caucasus (*H. bacescui*), and ER Y—Middle East [26–31].

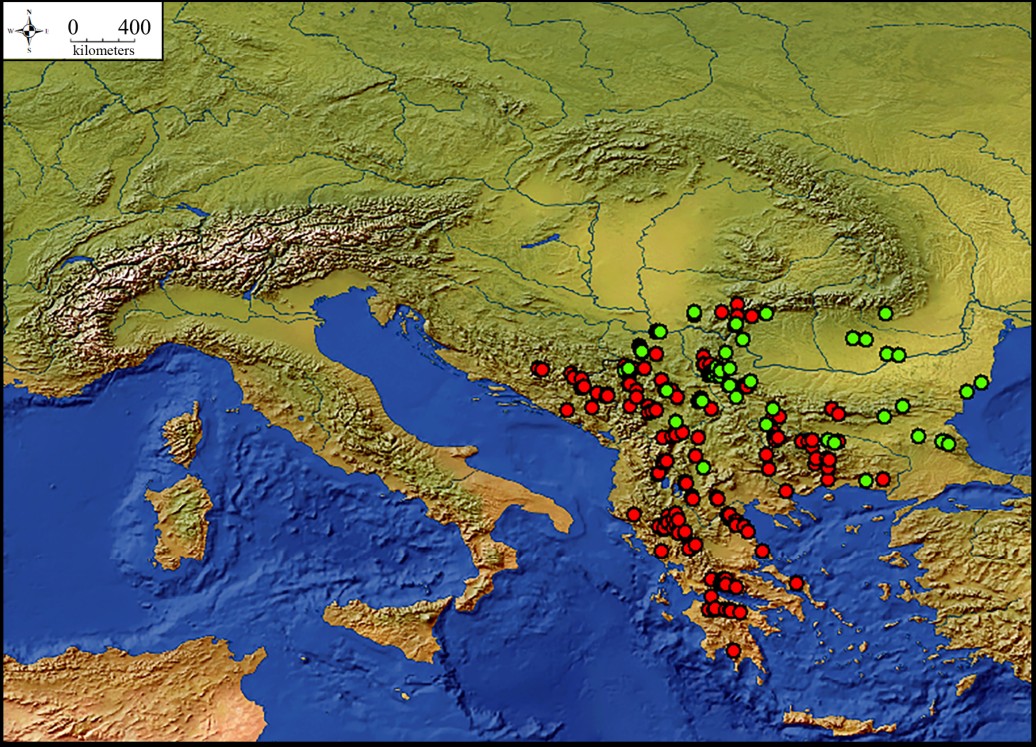

**Figure 1.** Current distribution data of *Helicopsyche bacescui* (green dots) and *Thremma anomalum* (red dots).

We selected these two caddisfly species because they exhibit some of the parameters that may be sensitive to climate change, such as preference for crenal and eucrenal zones, preference for cold water temperatures, and both species are grazers (specialists) and therefore have limited ecological niches [32]. By using SDMs and these two species as model organisms, we were able to: (i) examine the effects of climate change on two Trichoptera species; (ii) establish which areas in the Balkans (and neighboring countries) will be potentially lost and/or which ones can potentially be colonized by these species in the future; (iii) test the effectiveness of protected areas for the conservation of freshwater biota by investigating the representation of these species within the existing PA network in the Balkans, taking into account that both species are classified as endangered in Serbia (according to IUCN criteria) and are also strictly protected [33].

## 2. Materials and Methods

### 2.1. Sampling Methods, Preservation, and Occurrence Data

Larvae were collected by hand using tweezers or a Surber net (catchment area 300 cm$^2$, mesh size 250 μm), while adults were captured using entomological nets or UV light traps. The material was placed into plastic vials and preserved with 70% or 96% ethanol. In the laboratory, identification was performed according to the relevant literature [23–25]. Material sampled in Serbia during the last 25 years is deposited in the collection of the Institute of Zoology, University of Belgrade–Faculty of Biology. In order to include the largest part of the area in the analysis, additional records of two species were obtained from the literature, considering neighboring countries [34–41]. All occurrence points were georeferenced for the purposes of this study and presented in Table S1 (Table S1: Trichoptera occurrence points). Duplicate entries have been removed.

### 2.2. Environmental Variables

Current and future bioclimatic data and altitude layers were downloaded from the WorldClim database [42]. Future projections of bioclimatic predictors were derived from the Hadley Global Environment Model 2 Earth System configuration—HadGEM2-ES [43] and the Community Climate System Model—CCSM4 [44]. Four representative concentration pathways (RCPs) [45] were used for each model, namely, RCP 2.6, RCP 4.5, RCP 6.0, and RCP 8.5 for the 2050 (averaged for 2041–2060) and 2070 (averaged for 2061–2080) time periods. Representative concentration pathways were defined in the IPCC's 5th Assessment Report of 2014. They represent the possible trajectories for greenhouse gas emissions [46]: RCP2.6 (the minimum greenhouse gas emission scenario), RCP4.5 and RCP 6.0 (medium greenhouse gas emission scenarios), and RCP8.5 (the maximum greenhouse gas emission scenario). Slope and aspect were calculated from the elevation layer. Habitat variables were obtained from the Corine Land Cover (CLC) database [47]. All variables had a spatial resolution of 2.5 arc minutes.

All environmental variables were first subjected to the multicollinearity test with variance inflation factor (VIF) analysis in the R platform [48] using the usdm package [49]. Environmental variables with high multicollinearity were removed until all remaining variables had VIF values below 5. The remaining variables: Mean Diurnal Range, Temperature Seasonality, Mean Temperature of Wettest Quarter, Mean Temperature of Driest Quarter, Precipitation of Wettest Month, Precipitation Seasonality, Precipitation of Coldest Quarter, Elevation, Slope, and Corine Land Cover were included in the models. The procedure was performed separately for each species.

### 2.3. Modelling Procedure

We used the maxent function of the R package dismo [50] to build species distribution models (SDMs) for the current and future distributions of the selected species. MaxEnt (maximum entropy algorithm) is a machine learning method based on the principle of maximum entropy that uses only presence data to calculate potential species distributions. As one of the most popular and widely used SDM tools [51–53], MaxEnt combines environmen-

tal variables and species occurrence to approximate potential geographic distribution [54]. Default MaxEnt settings were used.

We used the ENMeval package [55] in R to manage model complexity and determine the optimal combination of MaxEnt feature classes and regularization multipliers. The feature classes (Table 1) encompass Linear features (L), Quadratic features (Q), Product features (P), Hinge features (H), and Threshold features (T), as well as eight specific values for the regularization multiplier (RM), ranging from 0.5 and increasing in increments of 0.5 to 4.0. In order to identify the most suitable model candidate, we evaluated a total of 48 models, encompassing all conceivable combinations of feature types (L, LQ, LQH, LQHP, LQHPT) and regularization multipliers. The selection process was guided by the Akaike Information Criterion (AICc), with particular emphasis on minimizing its values (ΔAICc = 0) [55,56]. The results we obtained from the best candidate model were then used to develop a model projection for two species. For each species, we developed one current and sixteen future projections of the potential species distribution. Delta AIC and kappa statistics were used to evaluate the prediction of the best candidate model. To evaluate the predictive performance of the models, we used TSS (the true skill statistic TSS), which is a good measure of accuracy [57]. TSS ranges from −1 to +1, with higher values of TSS indicating better model performance [58].

**Table 1.** Model comparison based on Akaike Information Criteria (AICc).

| Species | Feature Classes | Regularization Multiplier | TSS | AICc | Parameters | Kappa |
|---------|-----------------|---------------------------|-----|------|------------|-------|
| *T. anomalum* | LQHPT | 4 | 0.6734059 | 4231.249 | 28 | 0.619919 |
| *H. bacescui* | LQHPT | 3.5 | 0.7419354 | 2196.356 | 31 | 0.6402334 |

For each species, maps were generated for both current and future scenarios (for 2050 and 2070) using two future projections and four RCP trajectories. A threshold value of "maximum sensitivity plus specificity" was selected to define suitable/unsuitable areas for species. This threshold was used to reclassify our models into binary presence/absence maps and conduct further analyses as suggested by Liu et al. [59]. For each RCP scenario separately, we calculated changes in species distribution ranges by comparing the percentages of areas gained or lost under the conditions of the different climate change scenarios. All analyses were performed in ArcGIS, ver. 10.3.

### 2.4. Effectiveness of Protected Areas

Potential current and future species distribution maps and the World Protected Areas Database map (http://www.wdpa.org) (accessed on 20 January 2019) were overlaid for each species to assess the effectiveness of currently designated protected areas. Only protected areas in IUCN categories I–VI were considered.

We calculated the percentage of a species' currently postulated range that overlaps with PAs, as well as the percentage of PA overlap with the species' predicted range under conditions of different climate change scenarios. All analyses were performed in ArcGIS, ver. 10.3.

### 3. Results

The results of the ENMeval package suggest the use of the LQHPT trait for *T. anomalum* and *H. bacescui*. The obtained TSS values of 0.67 and 0.74 for *T. anomalum* and *H. bacescui*, respectively, indicate a good fit of the models (Table 1).

The relative importance of environmental variables varied among the studied species (Table 2). Slope contributed most to the distribution of *T. anomalum* (50.32%), followed by elevation (23.06%), while the potential distribution of *H. bacescui* was strongly influenced by precipitation of the wettest month (BIO13, 22.97%) and mean temperature of the wettest quarter (BIO8, 20.53%). The contributions of other climatic variables were less than 15% for both studied species.

**Table 2.** Percentage contributions of the studied variables to the modelled distribution of *T. anomalum* and *H. bacescui*.

| Environmental Variables | *H. bacescui* | *T. anomalum* |
|---|---|---|
| Mean Diurnal Range (BIO2) | - | −4.08 |
| Temperature Seasonality (BIO4) | −15.67 | - |
| Mean Temperature of Wettest Quarter (BIO8) | 20.53 | 2.84 |
| Mean Temperature of Driest Quarter (BIO9) | 13.13 | 4.08 |
| Precipitation of Wettest Month (BIO13) | 22.97 | - |
| Precipitation Seasonality (BIO15) | 13.06 | 1.21 |
| Precipitation of Coldest Quarter (BIO19) | - | 9.22 |
| Elevation | 11.92 | 23.06 |
| Slope | 6.45 | 50.32 |
| Corine Land Cover | 1.36 | 5.18 |

Our results showed that distributions of the species *T. anomalum* and *H. bacescui* will change in different proportions under conditions of all future climate change scenarios and models (Table 3). According to the CCSM4 and HadGEM2-ES results, *H. bacescui* will be negatively affected by climate change and is predicted to lose from 24.21% of its range (based on the CCSM4 prediction for 2070, RCP 6.0) up to 67.68% (according to the HadGEM2-ES forecast for 2050, RCP 4.5), mostly from the southern part of the investigated area (Table 3, Figure S1). Moreover, *H. bacescui* in the future may shift its range from the south to the north, mostly on the Carpathian mountains under conditions of both climate models (Figure S1). On the other hand, across all climate models and scenarios, by 2050 and 2070, *T. anomalum* may extend its range (by up to 71.14%) in the north and east on the Carpathian Mountains (Table 3, Figure S2).

**Table 3.** Percentage range reduction/expansion for the investigated species based on predictions of the HadGEM2-ES and CCSM4 climate models and four RCP scenarios.

| Species | Climate Model | RCP | 2050 | 2070 |
|---|---|---|---|---|
| *H. bacescui* | HadGEM2-ES | 2.6 | −56.69 | −45.83 |
| | | 4.5 | −67.68 | −49.56 |
| | | 6.0 | −40.51 | −47.58 |
| | | 8.5 | −36.82 | −31.10 |
| | CCSM4 | 2.6 | −39.07 | −24.46 |
| | | 4.5 | −27.97 | −25.31 |
| | | 6.0 | −40.50 | −24.21 |
| | | 8.5 | −31.59 | −38.39 |
| *T. anomalum* | HadGEM2-ES | 2.6 | 46.16 | 45.27 |
| | | 4.5 | 45.59 | 57.75 |
| | | 6.0 | 39.90 | 54.94 |
| | | 8.5 | 54.14 | 71.14 |
| | CCSM4 | 2.6 | 21.07 | 19.40 |
| | | 4.5 | 35.39 | 35.35 |
| | | 6.0 | 28.07 | 35.12 |
| | | 8.5 | 31.48 | 45.46 |

Our analyses have shown that currently established protected areas do not cover a significant portion of the potential current and future ranges of the studied species (Table 4, Figures S3 and S4).

**Table 4.** Percentage of nationally protected areas that overlap with projected species distributions.

| Climate Model | RCP | H. bacescui | | T. anomalum | |
|---|---|---|---|---|---|
| | | **2050** | **2070** | **2050** | **2070** |
| HadGEM2-ES | 2.6 | 5.28 | 7.81 | 25.91 | 25.60 |
| | 4.5 | 7.23 | 3.85 | 25.29 | 26.79 |
| | 6.0 | 6.62 | 4.00 | 24.41 | 26.38 |
| | 8.5 | 5.33 | 5.60 | 26.33 | 27.63 |
| CCSM4 | 2.6 | 5.56 | 7.45 | 22.26 | 21.65 |
| | 4.5 | 7.23 | 7.68 | 24.47 | 24.49 |
| | 6.0 | 6.62 | 8.94 | 23.81 | 24.62 |
| | 8.5 | 4.93 | 5.60 | 23.62 | 25.55 |

Current overlap for *H. bacescui* is 14.34% and for *T. anomalum,* it is 18.97%.

Less than 20% of the species' potential current range is in protected areas (18.97% for *T. anomalum* and 14.34% for *H. bacescui*). In the future, the percentage of projected areas within PAs will decrease (up to 9%) for *H. bacescui* under conditions of both models and scenarios compared to the current potential distribution. A higher percentage of projected areas in the future that overlap with PAs was recorded for *T. anomalum* in both climate models and time periods, especially in the Carpathian Mountains (Table 4, Figure S4).

## 4. Discussion

*Helicopsyche* von Siebold, 1856 and *Thremma* McLachlan, 1876 are genera of ancient origin, probably from the early Tertiary [60]. Five species of both genera are known in Europe [23–26], most of them endemic/microendemic to the Mediterranean region, suggesting their Mediterranean origin [60]. Although endemic species are considered more sensitive to climate change due to their overall limited dispersal [16,32], it has been challenging to assess the impact of climate change on more widespread species, such as the studied *H. bacescui* and *T. anomalum*. In this paper, we assessed the distribution of these two locally/regionally threatened Trichoptera species under conditions of two climate change models and four RCP scenarios for two time periods to estimate the effects of global warming on their distribution and to evaluate the effectiveness of PAs in covering their potential current and future ranges.

Our results showed that the most significant variables affecting the distribution of *H. bacescui* was precipitation of wettest month (BIO13, 22.97%) and mean temperature of wettest quarter (BIO8, 20.53%). Larvae of *H. bacescui* are adapted to life in small, shallow, forested lowland streams and brooks, with low water temperature and low flow velocity [27,36]. The hydrological regime of freshwaters is closely related to precipitation, so changes in precipitation dynamics can have direct or indirect effects on aquatic ecosystems and their inhabitants [8,61]. Late spring precipitation can increase flow velocity in streams, which affects the development of Trichoptera larvae [15]. Another negative effect of strong water flows on larvae is the destabilization of substrate and food resources [15]. For *T. anomalum*, our results showed that slope was the predictor that contributed most to the distribution pattern of this species. It has been shown that slope is an important factor affecting flow velocity and oxygen content [12], thus influencing the distribution of *T. anomalum*, which prefers zones of moderate to fast flow and is a typical inhabitant of reocrenal springs [28]. In Serbian waters, the most stable populations of *T. anomalum* were found in rheocrenal mountain springs on the substrate covered with moss [28].

We predict that the potential distributions of *H. bacescui* and *T. anomalum* will change to varying degrees under the conditions of all future climate change scenarios and models. Differences among models concerning projected range change in 2050 and 2070 for both species are small, but species ranges would be more reduced under the HADGEM2—ES climate model than under the CCSM4 climate model. *H. bacescui* will be negatively affected by climate change, possibly losing up to 68% of its range, mainly in the southern part of the study area, and shifting its range northward, mainly to the Carpathian Mountains. On the

other hand, *T. anomalum* could expand its range by up to 72% across all climate models and scenarios, mainly to the north and east, also in the Carpathians.

Although the two species live primarily in headwater streams, there are differences in the ecological preferences of their larvae, which could explain the different responses to climate change. The study by Živić et al. [28] indicated that the temperature at most localities where *T. anomalum* was found in Serbian waters ranged between 10 and 18 °C. Also, when monitoring water temperatures from spring to autumn in Greece, this species was recorded at temperatures between 5.2 °C and 16.8 °C [25,36]. These data suggest that *T. anomalum* is rather tolerant of temperature fluctuations. On the other hand, the mean water temperature in Serbian waters with *H. bacescui* records was $9.9 \pm 0.8$ °C [27]. The data contained in Williams [62], obtained through personal communication with Dr Hans Malicky, indicate that *H. bacescui* can only tolerate temperature fluctuations between 8 and 15 °C. The study by Leathers et al. [63] shows that warming scenarios under climate change will affect cold water-adapted invertebrates that can tolerate temperatures below 15 °C. Compared to *T. anomalum*, *H. bacescui* could be considered a true cold stenothermic species that is overall more sensitive to temperature fluctuations. This is highly important from the perspective of habitats preferred by *H. bacescui*, as small streams and brooks may be affected by high air temperatures due to the close relationship between water and air temperatures [8]. In the study concerning the vulnerability of British aquatic insects, it was concluded that caddisfly species found in small lowland streams are at risk from climate change [64].

*H. bacescui* is a species restricted to small, isolated, and sporadic populations with a rather patchy (discontinuous) distribution [27,31]. Species with restricted distributions and small populations are considered less adaptable to environmental changes [65]. *T. anomalum* has a wider distribution and more stable populations [28], while we can assume that *H. bacescui* is more habitat specific compared to *T. anomalum*, and the models in our analysis likely predicted the loss of such habitats in the climate change scenario. In addition, our previous studies documented an eastward range shift of *T. anomalum* for the entire western boundary [28], so this species may have greater potential to colonize new areas. Our models have shown that both studied species tend to shift their range northward, mainly into the Carpathian Mountains. However, a study on the distribution patterns and ecological preferences of European caddisflies [32] has shown that after the last ice age, mainly generalists and species with a high dispersal ability recolonized northern Europe. Whether species are able to colonize new areas outside their current range depends largely on geographic barriers and dispersal capacity. Unfortunately, there is limited information on the dispersal ability of caddisfly species [32]. According to the literature data, caddisflies can disperse well at the local scale, especially their larval stages, but the lateral flight of adults contributes most to the colonization of new watersheds [66]. During their flight, surrounding vegetation has a major influence on dispersal distance [67]. Overall, a complex interaction of abiotic and biotic factors determines the current and future distribution of species, which is difficult to predict for species with larger ranges and lower habitat specificity [68].

Our results suggest that currently established PAs are insufficient to cover the potential current and predicted future ranges of the studied species. Most studies that evaluated the effectiveness of PAs under climate change conditions focused on terrestrial species [69–71]. Perhaps the main reason is that most protected areas have been established primarily to protect vertebrates and plants in terrestrial ecosystems. However, freshwater ecosystems are among the most threatened ecosystems in terms of biodiversity loss [3,72], especially with respect to sensitive aquatic insect orders [64,73,74], and they support 9.5% of all described animal species [75]. Based on our models, *H. bacescui* will lose approximately one- to two-thirds of its range depending on climate model implementation. Although *T. anomalum* is predicted to expand its range, habitat protection remains low (up to 30%). In Serbia, both species are listed as "strictly protected" in national legislation [76] and classified as Endangered according to IUCN categorization [33]. However, as far as we

know, these species are not protected in any form in other countries, nor included in the IUCN Red List of Threatened Species. In northern Macedonia, a decline in populations of *T. anomalum* has been noted, which may be related to anthropogenic pressures, especially water extraction and spring capturing [40]. Together with such anthropogenic activities, climate change could be one of the threatening factors that may contribute to the loss of the species [77]. Therefore, it is important not only to implement the legal protection of the species themselves, but also to ensure the protection of their habitats in other countries where they occur. It is important to expand the boundaries of current protected areas or establish new ones to preserve the potential habitat of these species as much as possible. In addition, potential new areas where the species could migrate in the future may not be protected, although they could provide habitat for them, even if it is less suitable compared to their current distribution [69]. Protecting such an area would be an important step in protecting the species considered in this study. Considering the high level of endemism of caddisflies in the Balkan Peninsula [78], but also the other representatives of aquatic fauna (fishes, hydroboid snails, malacostracans) [79], this region is of particular interest to assess whether current protected areas have been adequately selected or should be expanded. Such species, even if not charismatic, could be treated as surrogate or umbrella species that provide protection to other species and their habitats. It should be kept in mind that stream ecosystems are among the most difficult to protect, as protected areas usually cover only parts of their catchment area [80]. Species distribution models can be a very useful tool to test the effectiveness of protected areas in conserving aquatic species that may be threatened by climate change, but also by strong anthropogenic pressures leading to the destruction of aquatic habitats and their surroundings. In addition, the implementation of the Natura 2000 network in Serbia and neighboring countries will help improve strategies for the conservation of stream fauna, and in conjunction with species distribution models, it will be possible to evaluate the effectiveness of protected areas, which is one of the most important tasks of conservation biology [22]. In conclusion, we emphasize that future designation of PAs needs to combine aquatic and terrestrial systems and include macroinvertebrates as a key group for sustaining biodiversity [81,82].

**Supplementary Materials:** The following supporting information can be downloaded at: https://www.mdpi.com/article/10.3390/d15090995/s1, Table S1: Trichoptera occurrence points; Figure S1: Modelled distributions of the species *H. bacescui* under current climate and future climatic models and all scenarios; Figure S2: Modelled distributions of the species *T. anomalum* under current climate and future climatic models and all scenarios; Figure S3: Projected climate suitability for the species *H. bacescui* in and outside protected areas; 1—protected areas without a projected species distribution, 2—potential species distributions in protected area, 3—absence of both a projected species distribution and protected area, 4—projected specieFIGUREs distribution outside of protected areas; Figure S4: Projected climate suitability for the species *T. anomalum* in and outside protected areas; 1—protected areas without a projected species distribution, 2—potential species distributions in protected area, 3—absence of both a projected species distribution and protected area, 4—projected species distribution outside of protected areas.

**Author Contributions:** Conceptualization, D.M., K.S. and M.M.; methodology, D.M. and M.M.; software, M.M. and M.R.P.; field investigation K.S. and I.Ž.; writing—original draft preparation, D.M. and K.S.; writing—review and editing, I.Ž., M.M. and M.R.P. All authors have read and agreed to the published version of the manuscript.

**Funding:** This work was financially supported by the Serbian Ministry of Science, Technological Development and Innovation (Grant Nos. 451-03-47/2023-01/200178, 451-03-47/2023-01/200125 and 451-03-47/2023-01/200358).

**Institutional Review Board Statement:** The study did not require ethical approval.

**Data Availability Statement:** Not applicable.

**Acknowledgments:** Many thanks to our colleagues Dalibor Stojanović and Miroslav Živić for their invaluable help in the field.

**Conflicts of Interest:** The authors declare no conflict of interest.

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
