# Peer review of "Destiny of Two Caddisfly Species under Global Climate Change"

_diversity, doi:10.3390/d15090995_

Round 1
Reviewer 1 Report
This paper examines the distributions of two caddisfly species and models their potential distributions in relation to climate change. The area of investigation is the Balkans and parts of adjacent countries. This is a valuable study given the important impacts of climate change on fauna and flora and particularly in this case, on aquatic insects. Furthermore, the study considers their actual and potential distributions in relation to protected areas and notes their inadequacy with respect to the study species. Significantly, areas established to protect particular species will also have broader value to biota in general, and I would like to see that point made more strongly with respect to stream faunas towards the end of the paper’s discussion.
The results obtained indicate that the two caddisflies will respond in different ways to climate change with one expanding its range and the other losing a substantial portion of its current range. The factors affecting them also differ in importance. An important finding is that temperature increase is not necessarily the driving force behind a change in range but that changes in precipitation patterns are also likely to have major effects in several ways.
The paper is well written in very good scientific English and I have only a few minor suggestions (below) in that regard. The Introduction is quite detailed but introduces the subject well, Methods are clear and the results are presented concisely. The discussion is very good and strongly referenced as is the introduction.
My main suggestions are:
1. To include an outline map of the study area showing the borders of the various countries making up the maps in Figures 1 and 2, and including the location of the Carpathian Mountains. Such a map will be appreciated by many readers who will not be familiar with the Balkan region.
2. I would like to see a table included in the text (not as a Supplementary table) defining the feature types (L, LQ, LQH, LQHP, LQHPT). Again, this will help readers who do not have modelling experience.
Other suggestions
Title: Change caddisflies to caddisfly
Abstract
Line 13. tend rather than seem
Line 21. Presumably the ranges shift to the Carpathians from lower altitude regions? Clarify.
Line 81. Insert “in” before mainly
Line 83. I am not familiar with the terms mesolithal, macrolithal and megalithal. Perhaps you could say cobble, large stone and boulders, terms all stream ecologists will know?
Line 96. Are these areas that will be lost to the two caddis species?
Line 104. 300 cm2. Superscript 2.
Line 114. Layers (plural)
Line 184-5. “and models (Table 2) ...and is...of its...”
Line 193. Table 2. Percentage (not Percentages)
Table 2. RCP = Representative Concentration Pathways. What do the numbers in the RCP column mean?
Line 219. I think you mean the Mediterranean region not continent.
Line 252. Simpler wording here would be, “Although the two species live primarily in headwater streams...which could explain their different responses to climate change.”
Line 258. Tolerant of
Line 261. Insert Hans, or Dr Hans, before Malicky
Line 264. Delete ‘as’
Line 270. In this paragraph can you suggest any possible effects on adults of H. bacescui? You do so in the next paragraph with respect to T. anomalum. I assume the scenarios would likely apply to both species.
Line 312. “...of these species and others” See earlier comment about protected areas.
Author Response
Please, see the attachment.

Reviewer 2 Report
Species distribution models (SDM) always seem a rather uncertain way to define the distribution of a freshwater taxon. The climatic variables in the models never seem likely to have a strong connection with the target species because they mostly deal with air temperatures or rainfall, which can only have fairly indirect effects on an aquatic animal. However, in this case the two headwater stream Trichoptera seem more likely than many to respond directly to changes in water temperature and discharge, variables which probably vary rapidly with air temperature and rain in small streams. Thus I think the results from the SDMs are credible even if I am somewhat sceptical about the modelling.
The paper is well written and presented and my main concerns are some lack of clarity in the explanation of the modelling procedure and the fact that the maps are so small that little can be understood from them. I would recommend that the authors consider the following points when revising their paper:
p.1, line 41: it would be better to say ‘various levels of biological organization … and various spatial scales’.
p.2, lines 61-74: I think this paragraph is unnecessary as the information presented is well known to your potential audience. I would delete this section.
p.2, lines 83-84: micro, meso, macro and megalithal are jargon and should be avoided. It would be clearer to describe substrate preferences by using words such as gravel, pebbles, cobbles and boulders. The phrase ‘regarding …..gilds’ could be deleted without loss of meaning.
p.2, lines 86-89: here it would be very useful to have a map showing the current distribution of both species. The maps in Figs. 1 and 2 showing current distribution are too small to be readily understood. They also need to show the study area in relation to the rest of Europe for those who are unfamiliar with the geography of the Balkans.
p.3, line 110: you do not need 16 references to refer to additional distribution records. Just give a couple of the most important.
p.3, line 143: unless you explain what ‘feature types’ and ‘regularisation multipliers’ are then most readers will be puzzled. It would probably be best to delete these details unless you can explain them simply.
p.6, Fig. 1: I don’t think this figure reveals much that has not already been revealed by Table 2. For each species the maps are generally very similar for each combination of projected year and climate model. If you want to retain the maps place them in an appendix. The same comments apply to the maps in Fig.2 and the data in Table 3.
p.6, Table 3: the line immediately beneath the species names is confusing. Place the data about the current percentage overlap in the caption to the table.
p.8, lines 234-236: what evidence do you have that increases in flow velocity affect larval development? Are there any studies which show this? I also doubt that high velocities will completely eliminate the periphyton that both species eat. High velocities may well reduce periphyton on rocks but not eliminate it.
Author Response
Please, see the attachment.

Reviewer 3 Report
The manuscript is a nice piece to improve our comphreension on impact of global climate change, specially in freshwater ecosystems. The content is present and discussed in a good level. The methods are appropriated and wll-explained. At end, the manuscript open new ways to delimit protection areas or improve them. Thus, I support the publication of the manuscripto by Diversity journal.
Author Response
Dear Reviewer,
We sincerely appreciate your invaluable support in facilitating the publication of our manuscript in the esteemed journal, Diversity. We have diligently strived to incorporate all pertinent references within the Introduction section, aiming to maintain its conciseness and clarity. We genuinely believe that the inclusion of further references might potentially burden the introductory part. Your comprehension of our perspective is immensely valued. Thank you for your kind consideration.
Sincerely,
Katarina Stojanović
Reviewer 4 Report
Congratutation. A good article and argument against "flagship thinking" and towards a broader incorporation of aquatics in conservation issues.
I have no critical remarks.
Author Response
Dear Reviewer,
We extend our sincere appreciation for your positive evaluation of our manuscript. Your support and appreciation means a lot to us. We are very pleased that our work has found favour with you and that you have not found any critical comments. Your feedback strengthens our confidence in the quality of our research, and we are grateful for your time and attention. With sincere appreciation, we thank you once again.
Best regards,
Katarina Stojanović
Round 2
Reviewer 2 Report
The authors have made all the revisions I suggested in my first review of this paper. I have two comments on the current version:
Fig.1: the scale on the map does not seem correct.
line 307: delete 'but overall stream biota'.
Author Response
Dear reviewer,
We have made the corrections. We have changed the scale and deleted the statement in line 307.
Best regards,
Katarina Stojanović